# Regulatory context drives conservation of glycine riboswitch aptamers

Matt Crum, Nikhil Ram-Mohan¤, Michelle M. Meyer *

Department of Biology, Boston College, Chestnut Hill, Massachusetts, United States of America

¤ Current address: Department of Emergency Medicine, Stanford University, Stanford, California, United States of America

* m.meyer@bc.edu

**Data Availability Statement:** All data was gathered from publicly available data and accessions for sequence used in analysis are provided within supporting information.

## Abstract

In comparison to protein coding sequences, the impact of mutation and natural selection on the sequence and function of non-coding (ncRNA) genes is not well understood. Many ncRNA genes are narrowly distributed to only a few organisms, and appear to be rapidly evolving. Compared to protein coding sequences, there are many challenges associated with assessment of ncRNAs that are not well addressed by conventional phylogenetic approaches, including: short sequence length, lack of primary sequence conservation, and the importance of secondary structure for biological function. Riboswitches are structured ncRNAs that directly interact with small molecules to regulate gene expression in bacteria. They typically consist of a ligand-binding domain (aptamer) whose folding changes drive changes in gene expression. The glycine riboswitch is among the most well-studied due to the widespread occurrence of a tandem aptamer arrangement (tandem), wherein two homologous aptamers interact with glycine and each other to regulate gene expression. However, a significant proportion of glycine riboswitches are comprised of single aptamers (singleton). Here we use graph clustering to circumvent the limitations of traditional phylogenetic analysis when studying the relationship between the tandem and singleton glycine aptamers. Graph clustering enables a broader range of pairwise comparison measures to be used to assess aptamer similarity. Using this approach, we show that one aptamer of the tandem glycine riboswitch pair is typically much more highly conserved, and that which aptamer is conserved depends on the regulated gene. Furthermore, our analysis also reveals that singleton aptamers are more similar to either the first or second tandem aptamer, again based on the regulated gene. Taken together, our findings suggest that tandem glycine riboswitches degrade into functional singletons, with the regulated gene(s) dictating which glycine-binding aptamer is conserved.

## Author summary

The glycine riboswitch is a ncRNA responsible for the regulation of several distinct gene sets in bacteria that is found with either one (singleton) or two (tandem) aptamers, each of which directly senses glycine. Which aptamer is more important for gene-regulation,

**Funding:** This work is partially funded by the National Science Foundation: MCB 1715440 (https://www.nsf.gov/), and the National Institutes of Health: GM115931 (https://www.nih.gov/). The funders had no role in the study design, data analysis, decision to publish, or preparation of the manuscript.

**Competing interests:** The authors have declared that no competing interests exist.

and the functional difference between tandem and singleton aptamers, are long-standing questions in the riboswitch field. Like many biologically functional RNAs, glycine aptamers require a specific 3D folded conformation. Thus, they have low primary sequence similarity across distantly related homologs, and large changes in sequence length that make creation and analysis of accurate multiple sequence alignments challenging. To better understand the relationship between tandem and singleton aptamers, we used a graph clustering approach that allows us to compare the similarity of aptamers using metrics that measure both sequence and structure similarity. Our investigation reveals that in tandem glycine riboswitches, one aptamer is more highly conserved than the other, and which aptamer is conserved depends on what gene(s) are regulated. Moreover, we find that many singleton glycine riboswitches likely originate from tandem riboswitches in which the ligand-binding site of the non-conserved aptamer has degraded over time.

## Introduction

Structured RNA motifs play vital roles in all kingdoms of life. They are essential in protein translation [1,2], perform catalytic functions [3,4], and regulate gene expression [5,6]. Understanding the evolution and conservation of structured RNAs provides insight into the cellular processes in which they are involved. However, in comparison to protein coding sequences, the impact of mutation and natural selection on the sequence and function of structured RNAs is not well understood. Some RNAs such as the ribosomal RNA, and other large functional RNAs, are widely distributed across entire kingdoms. However, many others are narrowly distributed to only a few organisms, suggesting that their evolution may be quite rapid [7,8].

The ability of traditional phylogenetic approaches to investigate the evolution of structured RNAs is frequently inadequate due to a reliance on primary sequence information alone. Currently, only RAxML provides an algorithm for phylogenetic analysis that takes secondary structure into account, but it relies on a consensus secondary structure provided after sequence alignment [9]. Though this is an advantage over software that rely solely on the primary sequence, this still removes the potential for subtle structural similarities that might be present in the case of a pairwise assessment of structural similarity. Moreover, the antagonistic role that decreasing sequence length and increasing taxonomic diversity play on phylogenetic confidence limits the investigation of many structured RNA regulators, which tend to be short and have high primary sequence diversity [10].

To overcome these challenges, we utilized graph clustering, in which each vertex corresponds to an RNA sequence and edges are weighted based on a distance metric (e.g. RNA structural similarity, sequence similarity, or some combination of both), to assess the relative similarity of related RNAs. A graph-based clustering method was recently utilized to cluster contigs belonging to the same transcripts in *de novo* transcriptome analysis [11], but it lacks the structural component necessary for accurate clustering of structured ncRNA sequences. Other approaches for detection and clustering of structured RNAs provide robust methods for researchers to classify members of existing RNA families [12–19] and identify novel structured ncRNA [20–25]. However, these tools are fundamentally not designed to compare degrees of conservation across clusters of homologous structured RNAs.

To investigate motif divergence an analysis approach that provides fine-scale comparison of similarity within and between groups of homologous RNAs is necessary. Our approach allows any measure of pairwise similarity between two RNA sequences to be utilized for comparison and relative conservation between groups to be investigated. Thus, measures that

incorporate purely secondary structure information [26,27], purely sequence information [28,29], or a combination of both [30–34], may be utilized so that all available information can be captured. While this approach does not yield the same kind of inferences concerning the line of descent as traditional phylogenetic approaches, it does enable many more variables to be accounted for while assessing ncRNA similarity.

Regulatory RNAs have quickly become the largest class of functional RNAs. Yet, our understanding of their evolution lags that of the larger catalytic RNAs such as the ribosome. One class of regulatory RNAs that, similarly to the ribosome, take their function from a three-dimensional structure are riboswitches. Riboswitches are bacterial cis-regulatory elements that occur within the mRNA 5'-UTR and alter transcription attenuation or translation initiation of gene(s) directly downstream in response to a specific ligand [5,6]. This is accomplished by coordination between an aptamer domain, that binds the ligand, and an expression platform that translates this binding into downstream gene expression. Several riboswitches are identified across many phyla of bacteria [6,35–39], and may be ancient in origin [40,41]. However, riboswitches have also evolved under varied environmental and genetic contexts. Typically aptamer domains are structurally well-conserved while expression platforms may be quite variable across different bacterial species with *Firmicutes* and *Fusobacteria* species showing a preference for transcription attenuation, and *Proteobacteria* and *Actinobacteria* species preferring translation inhibition mechanisms [6].

The wide-spread distribution and structurally well-conserved aptamer domains of riboswitches make them useful models to better understand evolution of structured RNA motifs. The glycine riboswitch provides a unique opportunity to directly assess how riboswitch architecture may change over time or be influenced by which genes are regulated. The glycine riboswitch is commonly found in a tandem conformation where two homologous aptamers interact through tertiary contacts to regulate a single expression platform (tandem) [42]. A more conventional single-aptamer conformation also appears in nature (singleton), but singleton glycine riboswitches require a "ghost-aptamer" that functions as a scaffold for tertiary interactions similar to those observed in tandem glycine riboswitches [43]. Glycine riboswitch singletons are divided into two types distinguished by the location of the ghost-aptamer with respect to the ligand-binding aptamer. Type-1 singletons have a ghost-aptamer 3' of the aptamer, while the ghost aptamer is 5' of the glycine aptamer for type-2 singletons [43]. The relationship between singleton and tandem glycine riboswitches is not well characterized, and the how and why of tandem vs singleton riboswitch emergence and conservation is a subject of debate. Glycine riboswitches have been identified regulating several different sets of genes (genomic context) [42,44–46], and may function as either expressional activators (On-switch) [42] or repressors (Off-switch) [46].

In order to assess the relationship between singleton and tandem glycine riboswitches we used both traditional phylogenetics and graph clustering approaches to examine glycine aptamer sequences across a range of diverse bacterial species. Our investigation reveals that genomic context effects which tandem glycine-binding aptamer is more highly conserved. It also demonstrates that singleton riboswitches are more similar to the first or second tandem aptamer based on genomic context. Taken together, our findings suggest strongly that many singleton glycine riboswitches result from degradation of tandems, with the genomic context dictating which glycine-binding aptamer retains ligand responsiveness.

## Results

### Glycine riboswitches within the *Bacillaceae* and *Vibronaceae* families cluster based on genomic context

The tandem glycine riboswitch conformation is well-studied biophysically [44,45,47–51]. However, there is lack of consensus regarding the mechanism of ligand-binding or which of

the tandem aptamers is more essential for ligand-binding to induce gene regulation. Extensive *in vitro* investigation of a tandem glycine riboswitch originating from *Vibrio cholerae* found that ligand binding of the second aptamer (aptamer-2) controlled the expression platform and gene expression, while the first aptamer (aptamer-1) primarily played a role in structural stabilization and aptamer dimerization [44]. However, *in vivo* investigation of a tandem glycine riboswitch within *Bacillus subtilis* found that disruption of aptamer-1's binding pocket impeded riboswitch regulation more strongly than disrupting aptamer-2's [45]. To resolve the differences observed between the *V. cholerae* and B. *subtilis* tandem riboswitches, we conducted a comprehensive sequence analysis of glycine aptamers.

To identify glycine riboswitch aptamers, we used the RFAM covariance model RF00504 to search RefSeq77 [14,15,52]. Identified aptamers within 100 nucleotides (nts) of each other were considered to be part of a tandem riboswitch. A tandem aptamer covariance model was created using infernal and trained from this identified set and used to search RefSeq77 to supplement the dataset [12,13]. In total, 2,998 individual riboswitches were identified, 2,216 tandem riboswitches and 782 singleton riboswitches. Each was classified by genomic context based on the RefSeq annotated function of the putatively regulated gene. This dataset does not include the variant glycine riboswitches identified in a previous study [53], as the vast majority of these examples are present in metagenomic data, and therefore are not within RefSeq77.

To determine whether the functional differences observed between the glycine riboswitches from *B. subtilis* and *V. cholerae* are reflective of detectable sequence variation/divergence across their respective families, we first performed a phylogenetic analysis on examples from our dataset found within the *Bacillaceae* and *Vibrionaceae* bacterial families. We gathered sequences spanning both aptamers from 48 *Bacillaceae* and 37 *Vibrionaceae* tandem riboswitches. Within this set, all 37 *Vibrionaceae* riboswitches regulate transport proteins (TP), while 41 *Bacillaceae* riboswitches regulate glycine cleavage system (GCV) and the remaining 7 regulate TP (**S1 Table**). We then utilized RAxML to generate a phylogenetic tree (**Fig 1A**) from these sequences and the consensus secondary structure from alignment to our tandem covariance model using Infernal. The tree shows clustering within the group of *Bacillaceae* riboswitches regulating GCV, as well as within *Bacillaceae* riboswitches regulating TPs. However, there is a clear phylogenetic separation of the two groups of *Bacillaceae* riboswitches, splitting them into distinct clades. Furthermore, the clade representing *Bacillaceae* riboswitches regulating TP more closely clustered with the clade of riboswitches from *Vibrionaceae* regulating TP, although each set forms a distinct group. This finding suggests that genomic context may play a prominent role in the evolution of tandem glycine riboswitches.

### *Bacillaceae* tandem riboswitches display different patterns of aptamer conservation based on genomic context

To investigate whether the evolutionary pressure driving divergence of tandem glycine riboswitches regulating GCV and TP occurs evenly across both aptamers, or is specific to a single aptamer, we split tandem riboswitches within the *Bacillaceae* family into individual aptamers (**S2 and S3 Tables**). This provided us with two groups: one containing all aptamer-1's (first aptamer) and one containing all aptamer-2's (second aptamer). We then generated a phylogenetic tree for each set to determine whether the phylogenetic divergence seen within the riboswitch set is explained by variances within one specific aptamer or is present in both (**Fig 1B and 1C**). Both aptamer-1 and aptamer-2 sets display clear clustering based on genomic context. This indicates that divergence of tandem glycine riboswitches in differing genomic contexts cannot be fully explained by variation within the first or second

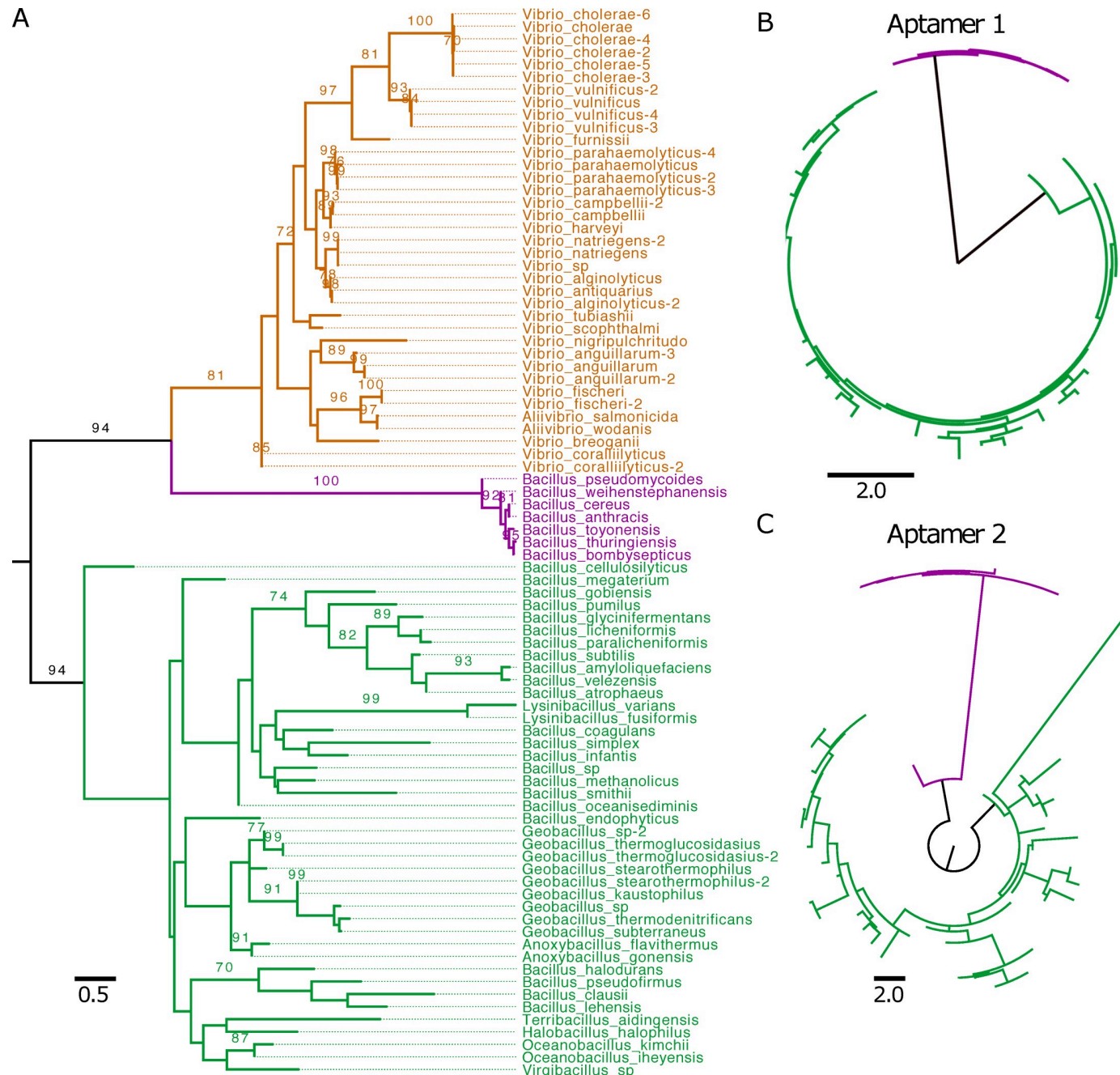

**Fig 1. Phylogenetic comparison of tandem riboswitches across Bacillaceae and Vibrionaceae.** A) 48 *Bacillaceae* and 37 *Vibrionaceae* tandem riboswitches were clustered based on aptamer sequence and structure across both aptamers of the riboswitch. After phylogenetic clustering, individual aptamers were colored based on the class of gene being regulated and the bacterial family of origin (*Vibrionaceae* TP are orange, *Bacillaceae* TP are purple, *Bacillaceae* GCV are green). Clusters have been labeled with the bacterial family and gene class being regulated. Bootstrap support values are displayed for 100 replicates when > = 70. B,C) Phylogenetic clustering of 48 tandem riboswitches, separated into aptamer-1 (B) and aptamer-2 (C), taken from the *Bacillaceae* family and colored according to class of gene being regulated (GCV are green, TP are purple). All trees are midpoint rooted.

aptamer alone. Moreover, within the GCV context, it appears that aptamer-1 is more highly conserved than aptamer-2, as indicated by the shorter branch lengths across the clade (**Fig 1B and 1C**).

## Genomic context dictates aptamer clustering in *Bacillaceae* and *Vibrionaceae*

To better understand how the homologous aptamers of the tandem glycine riboswitch have diverged, we broadened our taxonomic scope and focused our investigation onto the individual aptamer domains of the glycine riboswitch. However, the shorter sequence length of the individual glycine aptamers confounded our analysis. Thus, relative conservation of aptamer-1 to aptamer-2 in different genomic contexts was investigated using graph clustering of all *Bacillaceae* and *Vibrionaceae* aptamers within a given genomic context, excluding identical aptamer pairs coming from different strains of the same species. This set comprised of 84 pairs of aptamer-1 and aptamer-2 from *Bacillaceae* regulating GCV and 36 pairs from *Vibrionaceae* regulating TP (**S4 and S5 Tables**). The number of TP riboswitches was reduced by one in this analysis compared to the previous, as one of the riboswitches was no longer unique within the set when evaluating only the individual aptamer sequences.

We generated networks comprised of vertices corresponding to individual glycine riboswitch aptamers with edges weighted based on the pairwise RNAmountAlign distance score [34]. RNAmountAlign was chosen as the primary metric for edge-weighting in our work due to its implemented use of primary sequence information and ensemble mountain distance of secondary structure to generate a pairwise score more quickly and efficiently than other software. After weighting with RNAmountAlign, edges were trimmed if they were below a selected RNAmountAlign threshold, thus altering the topology of the network from completely pairwise to containing clusters of aptamers whose similarity is greater than the threshold. Thresholding was done across a range of RNAmountAlign scores to identify conserved aptamer groups which retained tight clustering (**Fig 2A**). Each network corresponds to a specific genomic context, TP or GCV. We find that within these contexts, aptamers group based on their position within the tandem arrangement (aptamer-1 vs. aptamer-2). Network density, defined as the fraction of edges present within a group compared to the total number of edges in the non-thresholded network, was calculated across a range of RNAmountAlign score thresholds and used to gauge relative conservation of each aptamer type for each genomic context (**Fig 2B**). Differing cluster densities between aptamer types revealed that genomic context effects which aptamer is more highly conserved: aptamer-1 is more highly conserved in riboswitches regulating the GCV, while aptamer-2 is more highly conserved in those regulating TP. A Wilcoxon rank-sum analysis of all intra-group edges was performed as well to validate these findings (**Fig 2C**). We obtain very similar findings using a variety of alternative distance metrics calculated using Dynalign [30,31], FoldAlign [32,33], Clustal Omega [28], and RNApdist [26,27] (**S1 and S2 Figs**).

## *Bacilli* class of bacteria shows clustering of singleton and tandem aptamers together

To assess the relationship between singleton and tandem riboswitches, we implemented graph clustering of individual aptamers from both tandem and singleton glycine riboswitches. We first categorized singleton aptamers within our dataset (includes all bacteria, refseq77-microbial) into singleton type-1 or singleton type-2 based on whether the ghost aptamer was found 3' or 5' of the glycine aptamer. Of 782 singleton riboswitches, 342 were characterized as singlet type-1, 125 as singleton type-2, and 305 were unable to be conclusively characterized as one or the other (designated singleton type-0) (**S6 Table**). We found that singleton type-1 riboswitches regulate GCV 93% of the time, while 90% of the singleton type-2 riboswitches regulate TP. This context dependent appearance of singleton riboswitches agrees with previous findings and gives confidence in our singleton annotation pipeline [54].

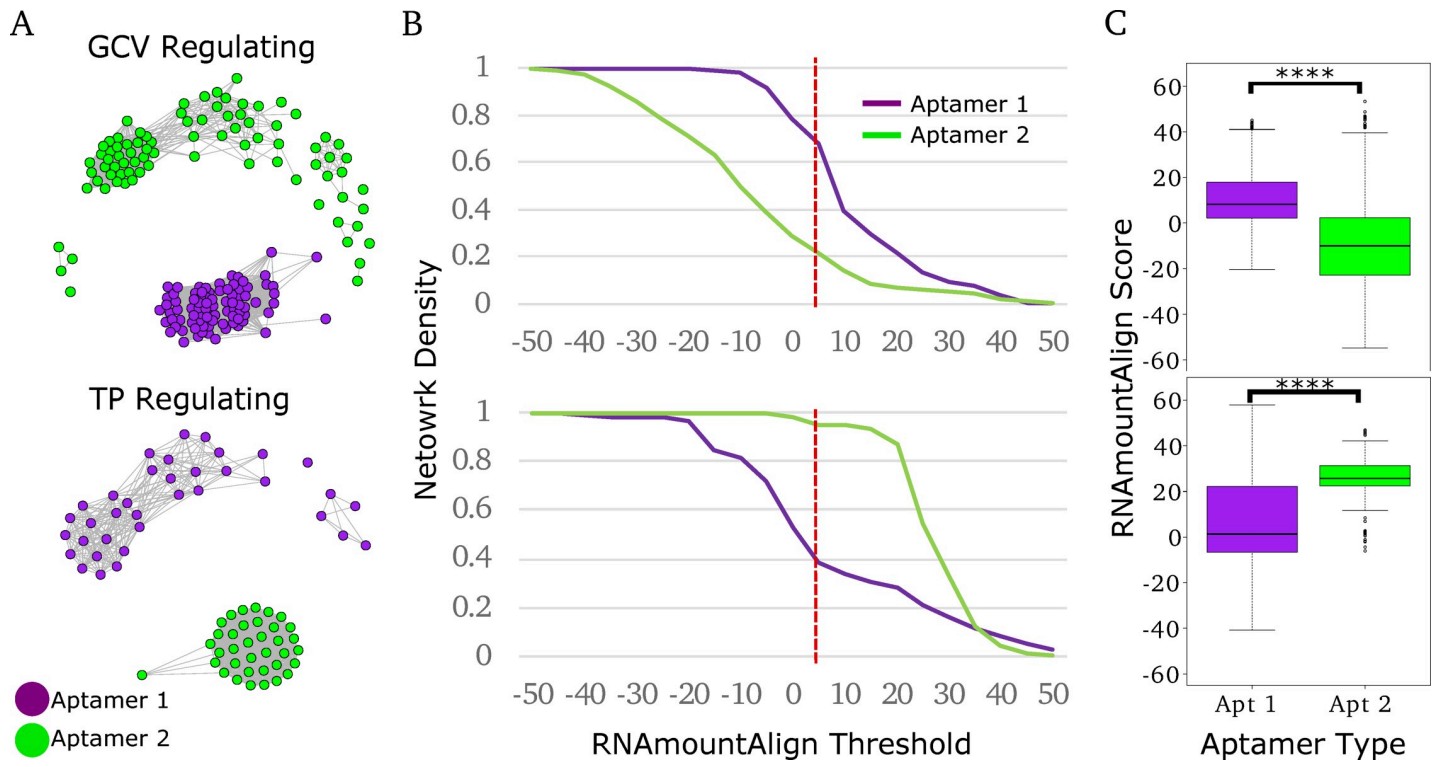

**Fig 2. Clustering of tandem riboswitch aptamers across Bacillaceae and Vibrionaceae.** A) *Bacillaceae* and *Vibrionaceae* tandem riboswitch aptamers were clustered using RNAmountAlign as a distance metric (threshold of 5). All represented *Bacillaceae* riboswitches regulate GCV (top), while *Vibrionaceae* riboswitches regulate TP (bottom). Aptamers are colored based on aptamer type, purple for aptamer-1 and green for aptamer-2. B) Network density was calculated for each aptamer in both networks across a range of RNAmountAlign thresholds. Dotted red line indicates the RNAmountAlign threshold (5) at which the networks in A were visualized. C) Box blots represent all pairwise edge-weights within each aptamer type. **** *p-value < 2x10^{-16}*.

We then implemented graph clustering on a set containing all glycine riboswitch aptamers from *Bacilli*, excluding identical aptamers coming from different strains of the same species, totaling 436 aptamers (**Fig 3A**) (**S7 Table**). This set from the *Bacilli* class was selected for its representation of both GCV (58%) and TP (31%) regulating riboswitches. Remaining riboswitches were labeled as regulating genes involved in glycine metabolism (Gly_Met) that is not part of the GCV operon or as Other. Using four distinct *de novo* community detection algorithms available in R (see methods) we identified modular communities within the set. Communities were selected based on each groups' core cluster, which was present within all community detection algorithms utilized. Aptamers that were found to be grouped with the core cluster in at least half of the community detection algorithms were subsequently added to the cluster. Cluster stability was verified using 100 replicates of parametric bootstrapping (**S3 Fig**) (see methods), as well as comparison to MCL [55] (**S4 Fig**) and DBSCAN [56] (**S5 Fig**) clustering output. This resulted in clusters comprised of a highly conserved core set and aptamers that closely grouped with them. Most communities contain either aptamers derived from a tandem arrangement or singleton aptamers. However, two communities included both singlet and tandem derived aptamers. The first contains singleton type-1 aptamers and aptamer-1 of tandem riboswitches, all regulating GCV. The second includes singleton type-2 aptamers and aptamer-2 of tandem riboswitches, all regulating TP (**Fig 3B**).

Members of both mixed communities were extracted and networks were generated for each as described above (**Fig 3C**). For aptamers originally part of a tandem arrangement, the paired aptamer was included to assess relative conservation (**S8 and S9 Tables**) of the singlet

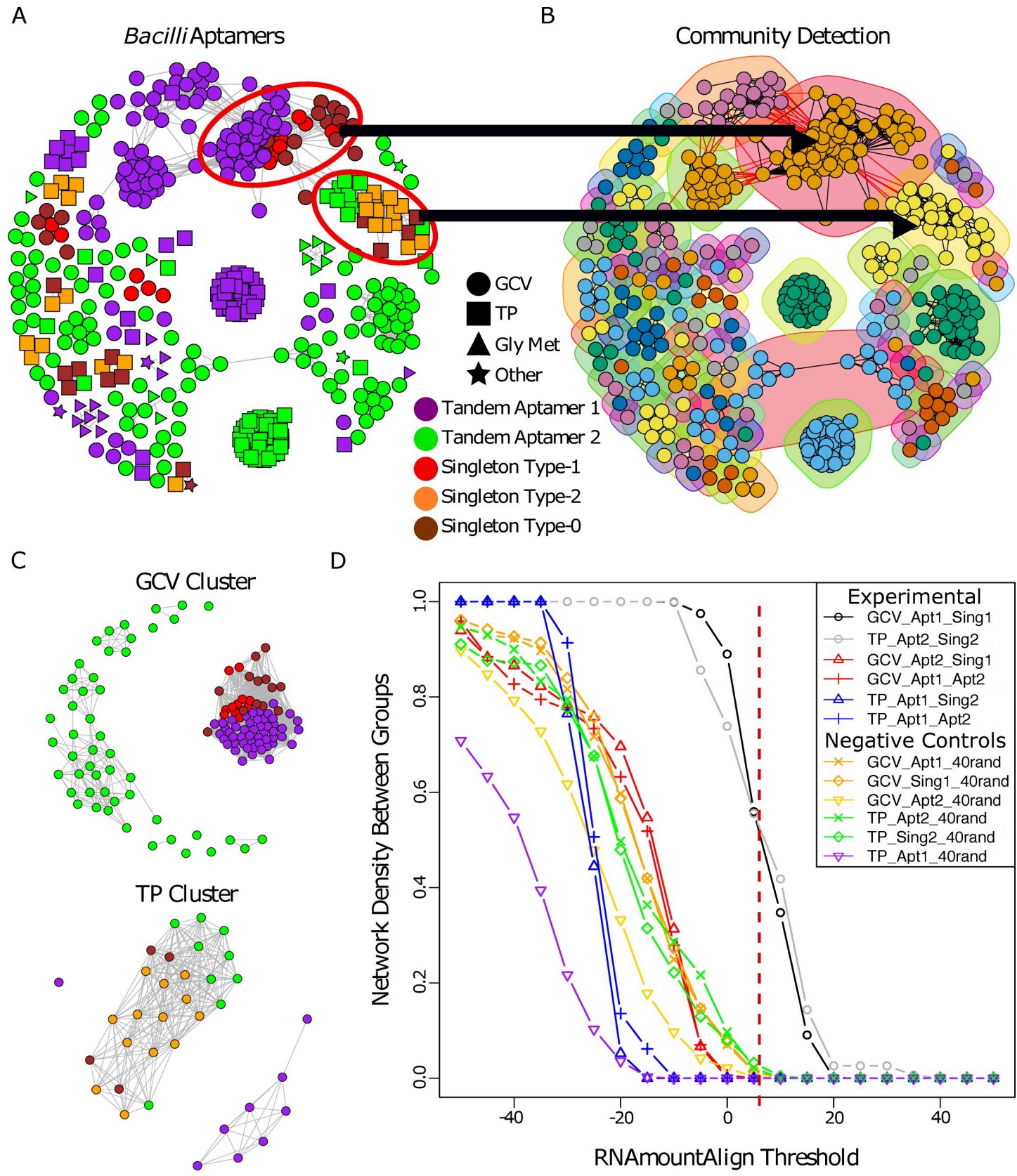

**Fig 3. Clustering of glycine riboswitch aptamers identified within the Bacilli class of bacteria.** A) Aptamers within the *Bacilli* bacterial class were identified and clustered based on RNAmountAlign pairwise similarity (visualized at threshold of 12). B) Sub-clusters (communities) were identified using the four community detection functions within R's igraph package. Two communities were identified that contain two different aptamer types: aptamer-1 and singlet type-1, and aptamer-2 and singlet type-2 that regulate GCV and TP respectively. Network shows visualization of the community detection algorithm cluster_fast_greedy (as implemented by R). Node colors correspond to distinct clusters detected. C) The two sub-clusters containing different aptamer types were parsed from the overall network, the tandem aptamers' partners were added to the set (as an out group within the same context), and graph clustering was visualized (RNAmountAlign threshold of 5). D) Edge density between aptamer groups was calculated for networks generated across a range of RNAmountAlign edge-weight thresholds. Dotted red line indicates the RNAmountAlign threshold (5) at which the networks in (C) were visualized.

aptamers to each tandem aptamer type. We determined relative conservation between aptamer types by calculating the network density of edges connecting each aptamer type (inter-edge density) (**Fig 3D**) across a range of RNAmountAlign thresholds. We observe that singleton type-1 aptamers regulating GCV are most similar to aptamer-1 of tandem riboswitches in the same context and conversely that singleton type-2 aptamers regulating TP are most similar to aptamer-2 of tandem riboswitches in the same context. The inter-edge density between singleton type-1 aptamers and tandem aptamer-1's regulating GCV is comparable to that seen between singleton type-2 aptamers and tandem aptamer-2's regulating TP (**Fig 3D**). These two groupings also represent the highest conservation across aptamer types within their networks, with other pairings being comparable to inter-edge density measurements with a random set of 40 aptamers (**S10 Table**). Using Dynalign, FoldAlign, Clustal Omega, and RNApdist as distance metrics yields similar findings (**S6** and **S7 Figs**).

To further investigate the similarities between these aptamers, we generated consensus structures of the riboswitches found within each genomic context using a combination of tools (see methods). Consensus structures of riboswitches regulating GCV show tandem aptamer-1 and singleton type-1 aptamers have high conservation of the P2 and P3 stems, as well as the binding pocket, while the P1 stem of tandem aptamer-2 shows high conservation with the singleton type-1 ghost aptamer (**Fig 4A and 4B**). This conservation of the ghost aptamer P1 stem correlates with the region required for tertiary interactions of tandem and singleton riboswitches [43]. This is observed within riboswitches regulating TP as well, except the aptamer of singleton type-2 and tandem aptamer-2 are the conserved aptamers (**Fig 4C and 4D**).

Together, our results indicate three things about *Bacilli* glycine riboswitch aptamers within each genomic context: 1) one tandem aptamer shows high conservation to the singleton aptamer, 2) conservation between the alternative tandem aptamer and singleton aptamers is no greater than conservation of the singleton to a random set of glycine riboswitch aptamers, and 3) ghost aptamer location correlates with the less conserved tandem aptamer. This fits with a model wherein these singleton riboswitches are the result of tandem riboswitch degradation, and which aptamer to be conserved and which to be degraded is dependent on genomic context. If the situation was reversed and tandems were the result of duplication events of singleton riboswitches, we would expect higher conservation between the singleton aptamer and both tandem aptamers compared to a random set of glycine riboswitch aptamers. However, we only observe such conservation with one tandem aptamer in each genomic context.

## *Actinobacteria* riboswitches display similar clustering pattern observed in *Bacilli*

To determine whether these patterns are observed within other clades of bacteria, we gathered all glycine aptamers within our dataset in the *Actinobacteria* phylum (distantly related to both the *Vibrio* and *Bacilli* classes analyzed previously), excluding identical aptamers coming from different strains of the same species, totaling 606 aptamers (**S11 Table**). We then evaluated all aptamers within the set in the same manner as our *Bacilli* investigation. Within this phylum,

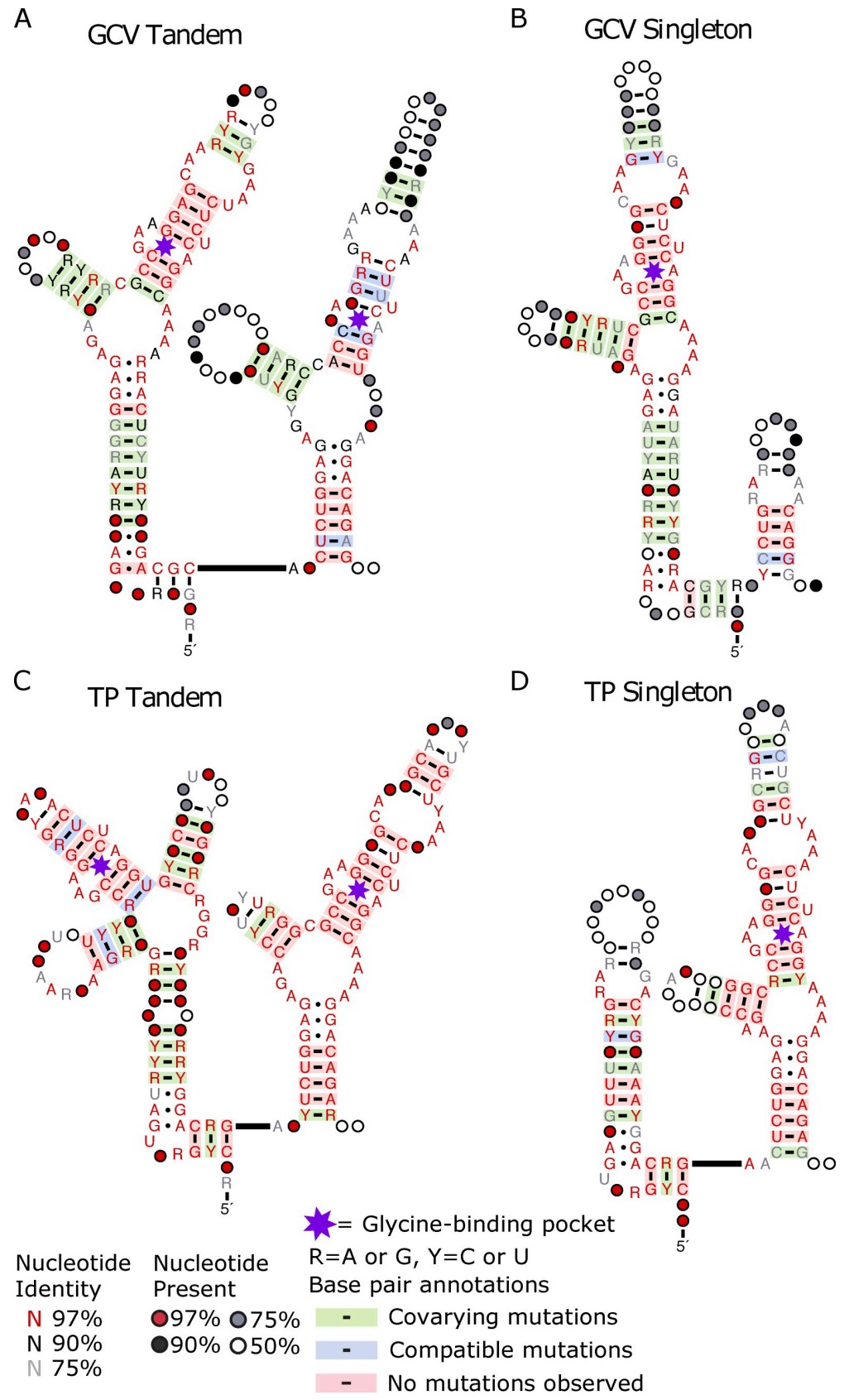

**Fig 4. Consensus structures of Bacilli riboswitches within a given genomic context display conservation between tandem and singleton aptamers.** Consensus secondary structure of the singleton and tandem riboswitches delineated by the genomic context. Conservation and covariation of base pairing generated using R2R with the individual covariance models. Tandem (A) and singleton (B) riboswitches regulating GCV. Tandem (C) and singleton (D) riboswitches regulating TP.

glycine riboswitches primarily regulate GCV (74%) or other genes involved in glycine metabolism (22%). We identified a group of 34 conserved aptamers corresponding to riboswitches regulating GCV (**S8A Fig**) and utilized *de novo* community detection algorithms to validate our observation (**S8B Fig**). Cluster stability was verified using 100 replicates of parametric bootstrapping (**S3 Fig**) (see methods), as well as comparison to MCL (**S9 Fig**) and DBSCAN (**S10 Fig**) clustering output. The aptamers within this group comprised primarily singleton type-1 aptamers and tandem aptamer-1 sequences, with five singleton type-0 and two singleton type-2 aptamer sequences accounting for the remainder. The singleton type-2 aptamers within the set may be misclassified aptamers or examples of singleton aptamers which do not conform to the patterns observed for other investigated aptamers. We performed graph clustering on the group, with paired tandem aptamer-2s included as an out-group, to investigate conservation of aptamer types (**S8C Fig**) (**S12 Table**). We then calculated the edge densities within and between singleton type-1 aptamers, tandem aptamer-1s, and tandem aptamer-2s, which demonstrate a clear conservation between singleton type-1 aptamers and tandem aptamer-1s (**S8D Fig**). These findings fit our conclusions drawn from the *Bacilli* class of bacteria. Using Dynalign, FoldAlign, Clustal Omega, and RNApdist as distance metrics yield similar findings (**S11 Fig**).

## Clustering based on genomic context is observed throughout entire bacterial kingdom

To determine whether clustering patterns observed within the *Bacilli* class and *Actinobacteria* phylum are reflected throughout the rest of the bacterial kingdom and can be observed among randomly selected aptamers, we randomly selected 150 distinct glycine riboswitch aptamers each for the GCV and TP genomic context (**S13** and **S14** **Tables**). Our selection retained comparable numbers of each aptamer type and excluded singleton type-0 aptamers. Singleton type-1 and type-2 aptamers are underrepresented in the TP and GCV regulating sets, respectively, because each aptamer type has few instances within that genomic context. Despite the diverse taxonomic range represented within this dataset, the generated networks display clustering patterns which align with our previous observations: a tendency towards clustering of singleton type-1 aptamers with tandem aptamer-1s when regulating GCV, and clustering of singleton type-2 aptamers with tandem aptamer-2s when regulating TP (**Fig 5**). Inter-edge density graphs of the aptamers shows similar trends to those seen within the *Bacilli* class and Actinobacteria phyla (**S12 Fig**). Using Dynalign, FoldAlign, Clustal Omega, and RNApdist as distance metrics yield similar findings (**S13** and **S14** **Figs**).

## Discussion

The tandem aptamers of the glycine riboswitch have fascinated RNA biologists since their identification in 2004 [42,57]. Extensive work has assessed whether the two homologous aptamers of the tandem glycine riboswitch functioned cooperatively [47–50], which tandem aptamer was more important for ligand binding [44,45], and what, if any, benefit a tandem conformation provided over the singleton glycine riboswitch [43]. In this work we use graph clustering analysis to investigate a similarly divisive question: what is the evolutionary

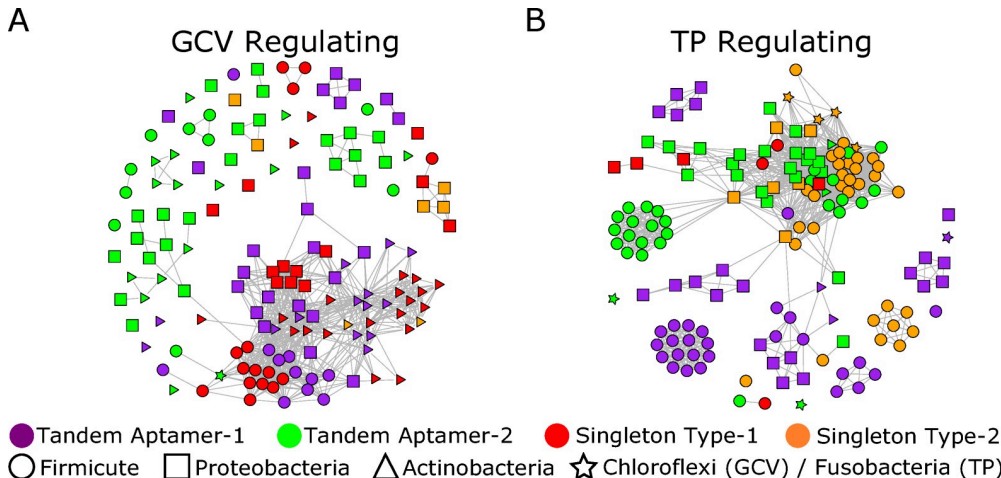

**Fig 5. Clustering of random glycine riboswitch aptamers across the bacterial kingdom.** A) Network visualization of 150 randomly chosen aptamers regulating GCV and clustered based on RNAmountAlign pairwise similarity (threshold -5). B) Network visualization of 150 randomly chosen aptamers regulating TP and clustered based on RNAmountAlign pairwise similarity (threshold -5). Only singletons that could be classified as type-1 or type-2 were included in this set.

relationship between tandem and singleton glycine riboswitches? While it may appear intuitive to believe that the tandem riboswitches that have been identified are the result of a duplication of identified singleton riboswitches, our findings point towards most singleton riboswitches being the result of tandem riboswitch degradation.

Phylogenetic evaluation of *Bacillaceae* and *Vibrionaceae* tandem riboswitches revealed that genomic context impacts riboswitch evolution. This is illustrated by *Bacillaceae* riboswitches which regulate TP grouping more closely with *Vibrionaceae* riboswitches regulating TP than with *Bacillaceae* riboswitches regulating GCV. Further investigation into the individual aptamers of *Bacillaceae* tandem riboswitches regulating GCV compared to TP showed that both aptamers individually show the same pattern of divergence.

Taking this analysis a step farther using graph clustering, we were able to determine that genomic context dictates which aptamer within a tandem glycine riboswitch is more highly conserved: aptamer-1 is more highly conserved in riboswitches regulating GCV, while aptamer-2 is more highly conserved in those regulating TP. These findings provide an elegant answer to a contradiction within the field in which investigations of diverse glycine riboswitch homologs yielded different results for whether ligand-binding of the first or second aptamer is more important for functionality [44,45]. Our results align with both studies' findings: the aptamer identified as the essential binding partner for regulation in each study is the aptamer found in our study to be more highly conserved within that genomic context. With the results of these previous studies combined with this new perspective provided by our data, it is reasonable to conclude that a difference in genomic context has driven glycine riboswitches to conserve different primary ligand-binding aptamers. Widespread horizontal transfer of the riboswitch with its accompanying gene could account for our findings. To investigate this possibility, we generated gene trees for aminomethyltransferases and symporters preceded by glycine riboswitches. From these trees, there is limited evidence of horizontal transfer of these genes (**S15 and S16** Figs) (**S15 and S16** Tables).

Our observation that tandem glycine riboswitch evolution is affected by genomic context led us to question the impact of genomic context on singleton glycine riboswitches. We extended our network analysis to singleton riboswitches, which provided valuable insight into

the relationship of tandem and singleton glycine riboswitches. Clustering of singleton and tandem aptamers from the *Bacilli* and *Actinobacteria* clades revealed that singleton aptamers are more similar to the first or second tandem aptamer based on genomic context: singleton type-1 aptamers regulating GCV are more similar to aptamer-1 of tandems regulating GCV, while singleton type-2 aptamers regulating TP are more similar to aptamer-2 of tandems regulating TP. This similarity of singletons to one tandem aptamer within a genomic context is highlighted by the fact that the singleton aptamers show no higher similarity with the other tandem aptamer than with a random set of 40 glycine riboswitch aptamers. This is observed within both GCV and TP regulating riboswitches and directs us towards the conclusion that singleton riboswitches are the remnants of degraded tandem aptamers.

We propose a model for the evolutionary path of the glycine riboswitch in which tandem riboswitches become singleton riboswitches by undergoing degradation of one aptamer into a ghost aptamer which retains regions relevant to tertiary interaction (**Fig 6**). In this model, the aptamer which is conserved is dependent on genomic context. The different conservation of tandem aptamers based on genomic context also fits recent studies that demonstrate a high likelihood that whether a glycine riboswitch is regulating TP or GCV is predictive of whether they are an on or off switch [42,45,46,54]. This fits a logical model for cellular response to high concentrations of glycine as a toxin [45,58–62]: genes responsible for glycine degradation become upregulated and those involved in glycine uptake become downregulated. In this way riboswitches in each genomic context protect the cell from the glycine toxicity as concentrations increase. This difference in regulation functionality accounts for riboswitches in different genomic contexts diverging, culminating in conservation of different aptamers and ultimately the formation of singleton riboswitches. It is possible that some singletons may have arisen from deletion of the middle section of a tandem riboswitch, leaving the 5' half of aptamer 1 and the 3' half of aptamer 2, resulting in a singleton. However, this scenario seems unlikely because it does not account for the ghost aptamer, which is important for structural stability of the glycine riboswitch.

While we have used graph clustering to unravel a specific discrepancy arising in existing experimental data, our approach may be used more broadly to assess how other riboswitches and other ncRNAs evolve and change over time. Variations in homologous riboswitch aptamers have demonstrated functional consequences. There are a range of variant riboswitch classes that interact with differing ligands [53,63,64], the most compelling of which are the homologous *ykkC* riboswitches [57,65] which include at least five subclasses each of which binds a distinct ligand involved in guanidine degradation and export [39,66–68]. There are also examples where structurally distinct riboswitches interact with the same or very similar ligands, such as the seven riboswitch classes involved in regulating S-adenosylmethionine concentration [69,70]. These RNAs include the SAM/SAH riboswitch which has been proposed to be a minimalistic form of a SAM riboswitch that evolved in organisms which readily degrade SAH [71]. The approaches we developed circumvent the limitations of traditional phylogenetic methods for assessing ncRNA similarity and enable identification of patterns in aptamer conservation that may point toward differences in biological function across diverse organisms.

## Materials and methods

### Riboswitch identification

Infernal's cmsearch function was used to query all RefSeq77 bacterial genomes using the RFAM glycine riboswitch covariance model to identify all individual putative glycine riboswitch aptamers (RF00504) [13,14,52]. All hits were filtered based on e-value, with a threshold of $1 \times 10^{-5}$. Putative riboswitches were sorted into components of a singleton or tandem glycine

## Model of Glycine Riboswitch Evolution

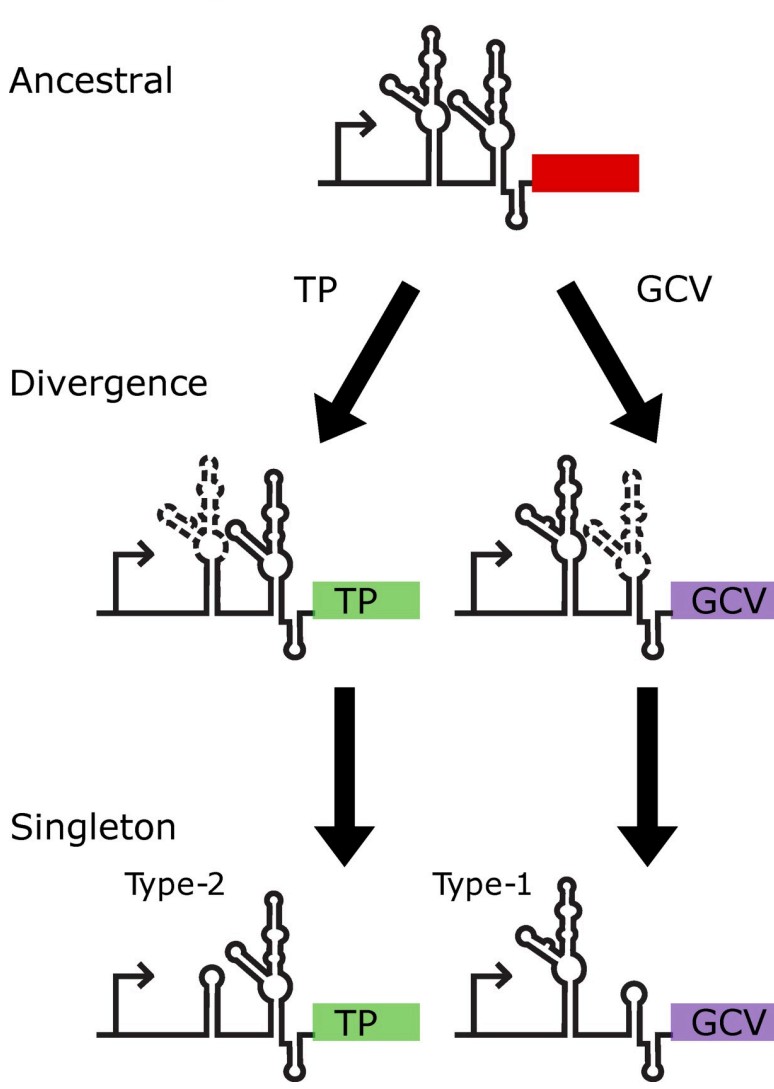

**Fig 6. Model of glycine riboswitch evolution.** Model proposed for the evolution and divergence of the glycine riboswitch. In this model a progenitor tandem riboswitch conserves one of the tandem aptamers based on the genomic context of the riboswitch, while the other slowly degrades down to the minimalistic components required for tertiary interaction to drive gene regulation. In this way, tandem glycine riboswitches may degrade into functional singleton tandem riboswitches.

riboswitch based on their proximity to any other putative aptamer. Two hits within 100nts of each other were considered to be the two aptamers of a tandem riboswitch; the largest distance between tandem aptamers observed was 32 nts. We then used a set of 30 tandem riboswitches to generate a covariance model which identified both tandem riboswitch aptamers together. The generated model is able to identify tandem glycine riboswitch aptamers, but does not explicitly include the expression platform due to the diversity of mechanisms of action for the glycine riboswitch. This tandem covariance model was used to query the RefSeq77 bacterial database and supplement our current set of putative glycine riboswitches with any tandems that may have been missed by the RFAM covariance model. 2,998 individual riboswitches were identified, 2,216 tandems riboswitches and 782 singleton riboswitches. Singletons were

then classified as type-1 or type-2 based on the location of the ghost aptamer, an adjacent stem structure that functions as a scaffold for tertiary interaction with the ligand-binding aptamer. Ghost aptamer location was determined based on conformation to covariance models generated from singleton type-1 and type-2 riboswitches reported in [54]. Of our 782 singleton riboswitches, 342 were characterized as singlet type-1, 125 were characterized as singleton type-2, and 305 were unable to be characterized as one or the other (called singleton type-0). Bedtools was utilized to determine the nearest downstream gene within 500 nucleotides on the same strand, providing a gene that is putatively regulated by each given riboswitch [72]. Genes were binned based on function for determination of genomic context of the glycine riboswitch.

### Riboswitch phylogenetic analysis

Tandem riboswitches were grouped based on taxonomic origin and genomic context. In order to incorporate secondary structure information, groups were aligned using LocARNA's mlocarna function for *de novo* alignment and folding [73–75] and Infernal's cmalign function to align to the tandem covariance model [12,13]. Maximum likelihood phylogenetic trees were generated from these aligned groups using RAxML [9]. Trees from alignments generated by cmalign were run with an accompanying secondary structure file to guide phylogenetic maximum likelihood analysis based on aptamer sequence and structure. In each case, 100 bootstrap replicates were performed and maximum likelihood bootstrap confidence values $>= 70$ are reported.

### Graph clustering and network generation

Graphs of aptamer sequences were generated, with vertices representing individual aptamers and edges representing a pairwise similarity metric relating aptamer pairs. Edge weights were then thresholded, resulting in trimmed networks of clustered aptamers containing only edges connecting pairs with higher similarity than the threshold value. Aptamer networks were generated to determine clustering based on a number of different pairwise metrics. These included partition function (RNApdist) [26,27], sequence and structural similarity (FoldAlign and Dynalign) [30–33], ensemble expected mountain height (RNAmountAlign) [34], and sequence similarity (Clustal Omega) [28]. Pairwise values were utilized to generate and visualize aptamer networks using the igraph and qgraph R-libraries [76,77]. Optimal visualization thresholds vary between sets relative to the taxonomic diversity represented within them. Clustering of riboswitch groups based on genomic context and aptamer type were compared by network density across a range of thresholds for each distance metric. Modular clusters were identified using igraph's community identification functions cluster_fast_greedy, cluster_walktrap, cluster_edge_betweenness, and cluster_leading_eigen.

Following cluster identification, we performed 100 replicates of a parametric bootstrapping analysis which perturbs 5% of the network and then re-clusters. This analysis perturbs the network by adding/removing edges at random in a 1:1 ratio, resulting in a network that contains the same nodes and an equivalent number of edges, but 5% of the edges connect different nodes. For each iteration, we determined the new clustering for our group of interest using the igraph community detection methods. These were then compared to the original group that had no perturbation. This comparison was done across all 100 iterations and uses Jaccard Similarity Index to calculate similarity of each post-perturbation cluster to the original cluster. Average Jaccard Similarity Index across the 100 iterations was used to determine robustness for our clusters (**S3 Fig**). Cluster composition was also validated by MCL [55] and DBSCAN [56] clustering.

Network density was calculated by determining the percent of total possible pairwise edges remaining for a given set of vertices after edge-trimming based on a distance metric threshold. Inter- and intra-group density calculations represent edge-density within a group and between groups, respectively. The network density measured across a range of thresholds correlates to aptamer similarity with respect to a given distance metric used to weight edges. These generated graphs equate to a flipped cumulative distribution of possible edges and actual edges for a cluster as we threshold the network based on edge weights.

## Consensus structure generation

We generated Stockholm files for sets of riboswitches using a combination of LocARNA's mlocarna function [73] and alignment to our covariance models using Infernal's cmsearch function [13]. Ralee was then used to perform minor curation of alignments and VARNA was implemented for secondary structure visualization throughout the process [78,79]. R2R was then used to generate consensus structures based on these Stockholm files [80]. The "#=GF R2R SetDrawingParam autoBreakPairs true" flag was used to allow for breaking of base pairs in instances where aptamer stems were not highly conserved.

## Supporting information

**S1 Fig. Clustering of Bacillaceae tandem riboswitch aptamers using Dynalign, FoldAlign, Clustal Omega, and RNApdist.** A) Dynalign intra-edge density across a range of -500 to 0 (x-axis reversed to display decreasing density).
B) FoldAlign intra-edge density across a range of 0 to 2000.
C) Clustal Omega intra-edge density across a range of 0 to 100.
D) RNApdist inter-edge density across a range of 0 to 100 (x-axis reversed to display decreasing density).
(EPS)

**S2 Fig. Clustering of Vibrionaceae tandem riboswitch aptamers using Dynalign, FoldAlign, Clustal Omega, and RNApdist.** A) Dynalign intra-edge density across a range of -500 to 0 (x-axis reversed to display decreasing density).
B) FoldAlign intra-edge density across a range of 0 to 2000.
C) Clustal Omega intra-edge density across a range of 0 to 100.
D) RNApdist inter-edge density across a range of 0 to 100 (x-axis reversed to display decreasing density).
(EPS)

**S3 Fig. Cluster stability after 100 bootstrap replicates for Bacilli and Actinobacteria clustering.** Average Jaccard Similarity Index after 100 bootstrap replicates for *Bacilli* cluster regulating GCV, *Bacilli* cluster regulating TP, and *Actinobacteria* cluster regulating GCV. The first row indicates the cluster and the first column indicates the community detection method used. The methods tend to show good cluster stability, particularly cluster_walktrap. However, the cluster_fast_greedy algorithm tended to over group clusters after bootstrapping, leading to poor Jaccard Similarity Indexes for some clusters.
(EPS)

**S4 Fig. MCL clustering of Bacilli riboswitch aptamers.** A) *Bacilli* riboswitch aptamers clustered based on RNAmountAlign pairwise similarity (visualized at threshold of 8). Node colors correspond to aptamer type and node shape corresponds to genomic context.
B) MCL clustering of *Bacilli* riboswitch aptamers, accomplished using R's MCL package. Nodes are colored to distinguish distinct clusters. Red circles correspond (roughly) to the set

of nodes used in our main analysis, as identified using four of R's igraph community detection functions.
(EPS)

**S5 Fig. DBSCAN clustering of Bacilli riboswitch aptamers.** A) *Bacilli* riboswitch aptamers clustered based on RNAmountAlign pairwise similarity (visualized at threshold of 12). Node colors correspond to aptamer type and node shape corresponds to genomic context.
B) DBSCAN clustering of *Bacilli* riboswitch aptamers, accomplished using R's fpc package. An epsilon value of 3.65 was used and the minimum neighbors was set 4. Nodes are colored to distinguish distinct clusters. Clustering of tandem aptamer 1 and singleton type-1 aptamers regulating GCV is observed in light blue.
C) DBSCAN clustering of *Bacilli* riboswitch aptamers, accomplished using R's fpc package. An epsilon value of 2.85 was used and the minimum neighbors was set 2. Nodes are colored to distinguish distinct clusters. Clustering of tandem aptamer 2 and singleton type-2 aptamers regulating TP is observed in yellow. Red circles correspond (roughly) to the set of nodes used in our main analysis, as identified using four of R's igraph community detection functions.
(EPS)

**S6 Fig. Clustering of Bacilli aptamer-1 and singleton type-1 aptamer subset using Dynalign, FoldAlign, Clustal Omega, and RNApdist.** A) Dynalign inter-edge density across a range of -500 to 0 (x-axis reversed to display decreasing density).
B) FoldAlign inter-edge density across a range of 0 to 2000.
C) Clustal Omega inter-edge density across a range of 0 to 100.
D) RNApdist inter-edge density across a range of 0 to 100 (x-axis reversed to display decreasing density).
(EPS)

**S7 Fig. Clustering of Bacilli aptamer-2 and singleton type-2 aptamer subset using Dynalign, FoldAlign, Clustal Omega, and RNApdist.** A) Dynalign inter-edge density across a range of -500 to 0 (x-axis reversed to display decreasing density).
B) FoldAlign inter-edge density across a range of 0 to 2000.
C) Clustal Omega inter-edge density across a range of 0 to 100.
D) RNApdist inter-edge density across a range of 0 to 100 (x-axis reversed to display decreasing density).
(EPS)

**S8 Fig. Clustering of glycine riboswitch aptamers identified within the Actinobacteria phylum of bacteria.** A) Aptamers within the *Actinobacteria* bacterial phylum were identified and clustered based on RNAmountAlign pairwise similarity (visualized at threshold of 12).
B) Sub-clusters (communities) were identified using the four community detection functions within R's igraph package. One community containing primarily two different aptamer types: aptamer-1 and singlet type-1 was identified. Display visualization uses the community detection algorithm cluster_fast_greedy. Node colors correspond to distinct clusters detected.
C) The community containing different aptamer types was parsed from the overall network, the tandem aptamers' partners were added (as an out group within the same context), and graph clustering was visualized (RNAmountAlign threshold of 5).
D) Edge density between aptamer groups was calculated for networks generated across a range of RNAmountAlign edge-weight thresholds. Solid lines correspond to edge density within a group and dashed line correspond to edge density between the two indicated groups. Dotted red line indicates the RNAmountAlign threshold (5) at which the networks in (C) were visualized.
(EPS)

**S9 Fig. MCL clustering of Actinobacteria riboswitch aptamers.** A) *Actinobacteria* riboswitch aptamers clustered based on RNAmountAlign pairwise similarity (visualized at threshold of 12). Node colors correspond to aptamer type and node shape corresponds to genomic context. B) MCL clustering of *Actinobacteria* riboswitch aptamers, accomplished using R's MCL package.

Red circles correspond (roughly) to the set of nodes used in our main analysis, as identified using four of R's igraph community detection functions.

(EPS)

**S10 Fig. DBSCAN clustering of Actinobacteria riboswitch aptamers.** A) *Actinobacteria* riboswitch aptamers clustered based on RNAmountAlign pairwise similarity (visualized at threshold of 12). Node colors correspond to aptamer type and node shape corresponds to genomic context.

B) DBSCAN clustering of *Actinobacteria* riboswitch aptamers, accomplished using R's fpc package. An epsilon value of 4 was used and the minimum neighbors was set 8. Nodes are colored to distinguish distinct clusters. Clustering of tandem aptamer 1 and singleton type-1 aptamers regulating GCV is observed in orange. Red circles correspond (roughly) to the set of nodes used in our main analysis, as identified using four of R's igraph community detection functions.

(EPS)

**S11 Fig. Clustering of Actinobacteria aptamer-1 and singleton type-1 aptamer subset using Dynalign, FoldAlign, Clustal Omega, and RNApdist.** A) Dynalign inter-edge density across a range of -500 to 0 (x-axis reversed to display decreasing density).

B) FoldAlign inter-edge density across a range of 0 to 2000.

C) Clustal Omega inter-edge density across a range of 0 to 100.

D) RNApdist inter-edge density across a range of 0 to 100 (x-axis reversed to display decreasing density).

(EPS)

**S12 Fig. Inter-edge density of random glycine riboswitch aptamer networks.** 150 aptamers regulating GCV and 150 aptamers regulating TP were randomly selected from across the bacterial kingdom and evaluated based on RNAmountAlign similarity score. Inter-edge density was calculated between aptamer types across a range of RNAmountAlign thresholds for the GCV regulating set (A) and the TP regulating set (B). Dotted red line on graphs indicate the threshold at which the clusters were visualized in Fig 5. Only singletons that could be classified as type-1 or type-2 were included in this set.

(EPS)

**S13 Fig. Clustering of random riboswitch aptamers regulating GCV using Dynalign, FoldAlign, Clustal Omega, and RNApdist.** A) Dynalign inter-edge density across a range of -500 to 0 (x-axis reversed to display decreasing density).

B) FoldAlign inter-edge density across a range of -500 to 1500.

C) Clustal Omega inter-edge density across a range of 0 to 100.

D) RNApdist inter-edge density across a range of 0 to 100 (x-axis reversed to display decreasing density).

(EPS)

**S14 Fig. Clustering of random riboswitch aptamers regulating TP using Dynalign, FoldAlign, Clustal Omega, and RNApdist.** A) Dynalign inter-edge density across a range of -500 to 0 (x-axis reversed to display decreasing density).

B) FoldAlign inter-edge density across a range of -500 to 1500.

C) Clustal Omega inter-edge density across a range of 0 to 100.

D) RNApdist inter-edge density across a range of 0 to 100 (x-axis reversed to display decreasing density).

(EPS)

**S15 Fig. Phylogenetic tree of genes coding for gcvT (aminomethyltransferase) and regulated by a glycine riboswitch.** 53 gcvT (aminomethyltransferase) genes regulated by glycine riboswitches were aligned using MUSCLE. RAxML was then used to generate a phylogenetic tree. Tips were then colored based on the phyla of bacteria the gene originates from (*Firmicutes* are red, *Proteobacteria* are blue). Bootstrap support values are displayed for 100 replicates when > = 70. B) All trees are midpoint rooted.

(EPS)

**S16 Fig. Phylogenetic tree of genes coding for sodium:amino-acid symporters and regulated by a glycine riboswitch.** 80 sodium:amino-acid symporter genes regulated by glycine riboswitches were aligned using MUSCLE. RAxML was then used to generate a phylogenetic tree. Tips were then colored based on the phyla of bacteria the gene originates from (*Firmicutes* are red, *Proteobacteria* are blue, *Actinobacteria* are green). Bootstrap support values are displayed for 100 replicates when > = 70. B) All trees are midpoint rooted.

(EPS)

**S1 Table. 48 *Bacillaceae and 37 Vibrionaceae tandem riboswitch sequences used for phylogenetic analysis.***

(XLSX)

**S2 Table. 48 *Bacillaceae tandem riboswitch aptamer-1 sequences used for phylogenetic analysis.***

(XLSX)

**S3 Table. 48 *Bacillaceae tandem riboswitch aptamer-2 sequences used for phylogenetic analysis.***

(XLSX)

**S4 Table. 168 *Bacillaceae tandem riboswitch aptamer sequences used for graph clustering analysis.***

(XLSX)

**S5 Table. 72 *Vibrionaceae tandem riboswitch aptamer sequences used for graph clustering analysis.***

(XLSX)

**S6 Table. 782 *glycine riboswitch singleton aptamer sequences labeled by type.***

(XLSX)

**S7 Table. 436 Bacilli riboswitch aptamer sequences for graph clustering analysis.**

(XLSX)

**S8 Table. 124 *Bacilli riboswitch aptamer sequences from aptamer-1 and singleton type-1 sub-cluster with paired aptamer-2 supplemented in.***

(XLSX)

**S9 Table. 35** *Bacilli riboswitch aptamer sequences from aptamer-2 and singleton type-2 sub-cluster with paired aptamer-1 supplemented in.*
(XLSX)

**S10 Table. 40 randomly selected aptamers used as out-group for inter-edge network density.**
(XLSX)

**S11 Table. 606** *Actinobacteria riboswitch aptamer sequences used for graph clustering analysis.*
(XLSX)

**S12 Table. 50** *Actinobacteria riboswitch aptamer sequences from aptamer-1 and singleton type-1 sub-cluster with paired aptamer-2 supplemented.*
(XLSX)

**S13 Table. 150** *riboswitch aptamer sequences regulating GCV for graph clustering analysis.*
(XLSX)

**S14 Table. 150 riboswitch aptamer sequences regulating TP for graph clustering analysis.**
(XLSX)

**S15 Table. 53** *gene sequences used for generating gcvT (aminomethyltransferase) gene tree.*
(XLSX)

**S16 Table. 80** *gene sequences used for generating sodium:amino-acid symporter gene tree.*
(XLSX)

## Acknowledgments

We want to thank Arianne Babina for essential insights into glycine riboswitch dynamics, Jon Anthony for groundwork on Infernal pipelines and servers, Daniel Beringer and Elizabeth Gray for assistance and feedback throughout the project, and Samantha Dyckman and Defne Surujon for support, proofreading, and brainstorming throughout the writing process.

## Author Contributions

**Conceptualization:** Michelle M. Meyer.

**Data curation:** Matt Crum.

**Funding acquisition:** Michelle M. Meyer.

**Investigation:** Matt Crum, Nikhil Ram-Mohan.

**Methodology:** Matt Crum, Nikhil Ram-Mohan.

**Supervision:** Michelle M. Meyer.

**Writing – original draft:** Matt Crum, Michelle M. Meyer.

**Writing – review & editing:** Matt Crum, Nikhil Ram-Mohan, Michelle M. Meyer.

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
