## [Decision Letter · Decision Letter 0]

7 Oct 2019

Dear Dr Meyer,

Thank you very much for submitting your manuscript, 'Regulatory context drives conservation of glycine riboswitch aptamers', to PLOS Computational Biology. As with all papers submitted to the journal, yours was fully evaluated by the PLOS Computational Biology editorial team, and in this case, by independent peer reviewers. The reviewers appreciated the attention to an important topic but identified some aspects of the manuscript that should be improved.

We would therefore like to ask you to modify the manuscript according to the review recommendations before we can consider your manuscript for acceptance. Your revisions should address the specific points made by each reviewer and we encourage you to respond to particular issues Please note while forming your response, if your article is accepted, you may have the opportunity to make the peer review history publicly available. The record will include editor decision letters (with reviews) and your responses to reviewer comments. If eligible, we will contact you to opt in or out.raised.

- Supporting Information uploaded as separate files, titled 'Dataset', 'Figure', 'Table', 'Text', 'Protocol', 'Audio', or 'Video'.

We hope to receive your revised manuscript within the next 30 days. If you anticipate any delay in its return, we ask that you let us know the expected resubmission date by email at ploscompbiol@plos.org.

Sincerely,

Shi-Jie Chen

Associate Editor

PLOS Computational Biology

William Noble

Deputy Editor

PLOS Computational Biology

[LINK]

Reviewer's Responses to Questions

**Comments to the Authors:**

Reviewer #1: Crum, Ram-Mohan and Meyer have explored the evolutionary relationships

between singleton and tandem glycine, using a number of different

approaches. This provides some evidence for which switch (5` or 3`) of

the tandem riboswitches is governing gene expression.

The number of phylogenetically informative sites in short RNAs is

generally very small, hence trees for these are often quite poor. With

doublet evolutionary models [1,2], using INDEL-aware evolutionary

models [3] and flanking sequence [4] being three strategies that have been

employed for addressing this problem.

The authors have used a number of approaches to ensure that their

results are robust, including more traditional methods (e.g. RaxML)

and non-traditional approaches such as RNAMountAlign. I am slightly

worried about the latter, if it's based upon the mountain metric

proposed by Moulten et al (2000), then there is an issue that the

outer-most basepairs in stems have too large an impact on the result,

furthermore the "correction" proposed by Moulton et al, fails to

correct the problem (based upon my own simulations). While this

shouldn't be a problem for this study, I suggesting treating this

method with caution.

MAJOR COMMENTS:

1. extensive use of graph clustering has been used, was bootstrapping or similar employed to ensure the clusters are robust? What about alternative clustering approaches e.g. MCL or DBSCAN?

2. If the tandem alignment/CM had been split into 5` and 3` halves, then the two half CMs could have also been used to determine which which half the singletons are likely derived from.

3. It is assumed that singletons are exclusively derived from either a 5` or 3` half of a tandem ancestor, yet half 5` + half 3` hybrids have not been excluded as a possibility.

E.g.

tandem 5555555555555555555lllll33333333333333333333

5p-single 5555555555555555555-------------------------

3p-single ------------------------33333333333333333333

5p3p-hybrid 5555555555------------------------3333333333

4. I didn't really understand the network density plots, are they flipped cumulative distributions?

5. Figures 2, 3 5 and 6 are very similar, are they all necessary?

MINOR COMMENTS:

1. what was the distribution of distances between the 2 halves of tandem models? This would help justify the "100 nucleotides" threshold.

2. Ralee is often used as well as R2R in order to curate alignments, was this also tried?

3. Were the GA thresholds provided by Rfam not useful, as opposed to an E-value threshold?

4. Were expression platforms included or excluded from the CM models?

5. Pg. 19, had forgotten what a "ghost" aptamer is by the time I encountered it again.

REFERENCES:

1. Schöniger M, von Haeseler A. Toward assigning helical regions in alignments of ribosomal RNA and testing the appropriateness of evolutionary models. Journal of molecular evolution. 1999 Nov 1;49(5):691-8.

2. Gesell T, Von Haeseler A. In silico sequence evolution with site-specific interactions along phylogenetic trees. Bioinformatics. 2005 Dec 6;22(6):716-22.

3. Rivas E, Eddy SR. Probabilistic phylogenetic inference with insertions and deletions. PLoS computational biology. 2008 Sep 19;4(9):e1000172.

4. Pignatelli M, Vilella AJ, Muffato M, Gordon L, White S, Flicek P, Herrero J. ncRNA orthologies in the vertebrate lineage. Database. 2016 Jan 1;2016. http://europepmc.org/articles/PMC4792531

Reviewer #2: review uploaded as an attachment

**Have all data underlying the figures and results presented in the manuscript been provided?**

Reviewer #1: Yes

Reviewer #2: Yes

PLOS authors have the option to publish the peer review history of their article (what does this mean?). If published, this will include your full peer review and any attached files.

Reviewer #1: Yes: Paul P. Gardner

Reviewer #2: No

---

## [Decision Letter · Decision Letter 1]

25 Nov 2019

Dear Dr Meyer,

We are pleased to inform you that your manuscript 'Regulatory context drives conservation of glycine riboswitch aptamers' has been provisionally accepted for publication in PLOS Computational Biology.

In the meantime, please log into Editorial Manager at https://www.editorialmanager.com/pcompbiol/, click the "Update My Information" link at the top of the page, and update your user information to ensure an efficient production and billing process.

One of the goals of PLOS is to make science accessible to educators and the public. PLOS staff issue occasional press releases and make early versions of PLOS Computational Biology articles available to science writers and journalists. PLOS staff also collaborate with Communication and Public Information Offices and would be happy to work with the relevant people at your institution or funding agency. If your institution or funding agency is interested in promoting your findings, please ask them to coordinate their releases with PLOS (contact ploscompbiol@plos.org).

Thank you again for supporting Open Access publishing. We look forward to publishing your paper in PLOS Computational Biology.

Sincerely,

Shi-Jie Chen

Associate Editor

PLOS Computational Biology

William Noble

Deputy Editor

PLOS Computational Biology

Reviewer's Responses to Questions

**Comments to the Authors:**

Reviewer #1: I am satisfied by the responses to my comments. Good job.

Reviewer #2: The authors have satisfactorily addressed all my queries and I therefore recommend that the manuscript be accepted for publication.

**Have all data underlying the figures and results presented in the manuscript been provided?**

Reviewer #1: Yes

Reviewer #2: Yes

PLOS authors have the option to publish the peer review history of their article (what does this mean?). If published, this will include your full peer review and any attached files.

Reviewer #1: No

Reviewer #2: No

---

## [Editor Report · Acceptance letter]

13 Dec 2019

PCOMPBIOL-D-19-01503R1 

Regulatory context drives conservation of glycine riboswitch aptamers

Dear Dr Meyer,

I am pleased to inform you that your manuscript has been formally accepted for publication in PLOS Computational Biology. Your manuscript is now with our production department and you will be notified of the publication date in due course.

With kind regards,

Matt Lyles
